# Effects of Foliar Application of ZnO Nanoparticles on Lentil Production, Stress Level and Nutritional Seed Quality under Field Conditions

**DOI:** 10.3390/nano12030310

**Published:** 2022-01-18

**Authors:** Marek Kolenčík, Dávid Ernst, Matej Komár, Martin Urík, Martin Šebesta, Ľuba Ďurišová, Marek Bujdoš, Ivan Černý, Juraj Chlpík, Martin Juriga, Ramakanth Illa, Yu Qian, Huan Feng, Gabriela Kratošová, Karla Čech Barabaszová, Ladislav Ducsay, Elena Aydın

**Affiliations:** 1Institute of Agronomic Sciences, Faculty of Agrobiology and Food Resources, Slovak University of Agriculture in Nitra, Tr. A. Hlinku 2, 949 76 Nitra, Slovakia; xkomarm@uniag.sk (M.K.); ivan.cerny@uniag.sk (I.Č.); juraj.chlpik@uniag.sk (J.C.); martin.juriga@uniag.sk (M.J.); ladislav.ducsay@uniag.sk (L.D.); 2Institute of Laboratory Research on Geomaterials, Faculty of Natural Sciences, Comenius University in Bratislava, Mlynska Dolina, Ilkovičová 6, 842 15 Bratislava, Slovakia; martin.urik@uniba.sk (M.U.); martin.sebesta@uniba.sk (M.Š.); marek.bujdos@uniba.sk (M.B.); 3Institute of Plant and Environmental Sciences, Faculty of Agrobiology and Food Resources, Slovak University of Agriculture in Nitra, Tr. A. Hlinku 2, 949 76 Nitra, Slovakia; luba.durisova@uniag.sk; 4Department of Chemistry, Rajiv Gandhi University of Knowledge Technologies, AP IIIT, Krishna District, Nuzvid 521202, India; ramakanthilla@yahoo.com; 5School of Ecology and Environmental Science, Yunnan University, 2 Cuihubei Lu, Kunming 650091, China; qianyu@ynu.edu.cn; 6Department of Earth and Environmental Studies, Montclair State University, 1 Normal Ave, Montclair, NJ 070 43, USA; fengh@montclair.edu; 7Nanotechnology Centre, CEET, VŠB Technical University of Ostrava, 17. listopadu 15/2172, 708 00 Ostrava-Poruba, Czech Republic; gabriela.kratosova@vsb.cz (G.K.); karla.cech.barabaszova@vsb.cz (K.Č.B.); 8Institute of Landscape Engineering, Faculty of Horticulture and Landscape Engineering, Slovak University of Agriculture in Nitra, Hospodárska 7, 949 76 Nitra, Slovakia; elena.aydin@uniag.sk

**Keywords:** lentil seeds, zinc oxide nanoparticles, nano-fertilizers, foliar application, physiological indexes, essential and beneficial nutrients

## Abstract

Nanotechnology offers new opportunities for the development of novel materials and strategies that improve technology and industry. This applies especially to agriculture, and our previous field studies have indicated that zinc oxide nanoparticles provide promising nano-fertilizer dispersion in sustainable agriculture. However, little is known about the precise ZnO-NP effects on legumes. Herein, 1 mg·L^−1^ ZnO-NP spray was dispersed on lentil plants to establish the direct NP effects on lentil production, seed nutritional quality, and stress response under field conditions. Although ZnO-NP exposure positively affected yield, thousand-seed weight and the number of pods per plant, there was no statistically significant difference in nutrient and anti-nutrient content in treated and untreated plant seeds. In contrast, the lentil water stress level was affected, and the stress response resulted in statistically significant changes in stomatal conductance, crop water stress index, and plant temperature. Foliar application of low ZnO-NP concentrations therefore proved promising in increasing crop production under field conditions, and this confirms ZnO-NP use as a viable strategy for sustainable agriculture.

## 1. Introduction

Legumes are nutritionally valuable and most important in the human diet by providing proteins, complex carbohydrates and dietary fiber [1,2]. Moreover, lentil legumes are especially high in proteins and essential amino acids, fiber and minerals and low in fat and cholesterol [1,3]. Lentil is an annual crop with moderate resistance to high temperatures, drought [4] and salinity [5], and it is acknowledged to have a long photoperiod [6]. Lentils are classified in the two main Chilean and Persian categories, dependent on seed size and weight. The seeds also come in various colors, and herein we used the black Lens culinaris ‘Beluga’ seeds in our field experiments because these provide an important source of beneficial nutrients [7].

Sufficient amounts of micronutrients are required for optimal lentil growth and development [8]. These include zinc (Zn) because its deficiency negatively affects plant reproductive functions and pollen development [9], reduces leaf area and induces chloroses and abnormal growth of plant structures. This deficiency can also result in higher sensitivity to biotic and abiotic stress [10]. Most importantly, Zn agronomic management includes foliar application and soil fertilization [11,12,13].

Zinc is also an essential micro-nutrient in general metabolism [13,14]. It forms a fundamental part of more than 300 enzymes [15], and it is involved in photosynthesis, DNA replication and transcription [13], hydrocarbon metabolism [16], regulation of auxin-production [17] and cell membrane integrity [13].

Macro-sized and soluble-Zn-ion fertilizers are traditionally applied in agriculture [18], and both ZnS and ZnO nanoparticles are promising novel fertilizer nutrients for crops [11,19,20,21].

The ZnO zincite mineral mostly occurs in wurtzite-type crystal symmetry which provides superior physical properties [22]. These include high electron mobility and wide band-gap energy, which both enable strong zincite photocatalytic reactivity [23,24]. Therefore, its morphological, spatial and structural forms in glass, electronics, energy, pigments, rubber textiles, cosmetics, food additives and pharmaceutics and medicine are complemented by its use in other technology [22,24,25].

ZnO-NP agricultural application provides great benefits for plant physiology and consequent production [11,12,20]. The physiological value is reflected in improved crop water stress index (CWSI), and in stomatal conductance (Ig), which is a function of plant water stress and leaf water potential level [26,27]. ZnO-NPs also influence plant temperature and temperature variations which are useful indicators of stress response, plant transpiration and energy balance [28].

However, currently available studies on ZnO-NP stress properties lack essential data. This especially applies to translocation, the metal-residues in plant tissues and their impact on seed quality and nutritional balance. This balance in fully ripe lentil seeds is important for higher phytic acid content, because this absorbs further micronutrients and contains phosphor as an anti-nutrient. Intestinal digestion further negatively affects nutrient bioavailability [29].

Although many experiments have investigated ZnO-NP and plant interaction during germination and early growth under greenhouse conditions [24,30,31], field lentil production critically depends on local climate and temperature [6]. Therefore, applied ZnO-NP field experiments are important to understand its effects on lentil production and physiological parameters. These affect all the following plant yield and thousand-seed weight, number of pods per plant, stomatal conductance, crop water stress index and additional temperature-derived lentil parameters. This lack of research inspired us to apply foliar ZnO-NPs to lentils under field conditions to acquire new information on the effects of ZnO-NP on these plants.

## 2. Materials and Methods

### 2.1. Plant Material

We chose the Lens culinaris ´Beluga´ legume. This variety originates in southwest Asia, is grown mainly in Canada [32] and has the third highest fiber and protein content of all legumes, with low-fat and low glycemic index [3,32,33]. Lens culinaris prefers a moderate climate with low humidity and long photoperiodic lighting, and does not require excess heat to thrive [34]. It adapts perfectly to loamy, well-drained and loose soils and those treated with compost before planting. The optimal soil pH for growth ranges from 6 to 6.5 [35].

### 2.2. Colloidal Properties of ZnO Nanoparticles

The ZnO-NP mineralogical specifications of size, morphology, crystal symmetry and unit cell dimensions have previously been reported [12]. Herein, we analyzed the colloidal properties of a solution prepared for foliar application. The hydrodynamic diameter and zeta-potential were measured by nanoPartica SZ-100 analyzer (Horiba, Kyoto, Japan). This was arranged with a microprocessor unit which directly calculated the zeta-potential at neutral pH and 25 °C laboratory conditions under the generalized Smoluchowski equation.

### 2.3. Experimental Site Description

Field experiments were performed in the Hostie Village near Zlaté Moravce in Slovakia. This lies 310 m above sea level in the north-eastern part of Žitavská pahorkatina Upland. The Pohronský Inovec Mts. lie to the east and the Nitrianska pahorkatina Wold and Tribeč Mts. are on the western aspect. Geomorphic classification indicates that the experimental field is in an upland area with rendzic leptosols and prior annual rotation of maize and potatoes. Soil availability of Zn, Cu, Mn and Fe was measured by DTPA extraction [36]. The content of soil carbonates, H_2_O and KCl pH, hydrolytic acidity, sum of exchangeable basic cations, sorption capacity, soil organic carbon and electrical conductivity were all measured as in Hrivňáková, et al. [37], and shown here in Table 1.

The detailed geographic, climate and soil conditions were previously reported by Kolenčík et al. [12], and the local monthly air temperature and precipitation are recorded from Meteoblue Web Weather Server. These are highlighted in Figure 1.

### 2.4. Field Experiment

The experiment was performed on 5 m^2^ parcels with silt loam soil [11]. Treatment and control plants were set in random orientation in the perpendicular blocks, and each had three parallel runs. Treatments lacking ZnO-NP foliar application formed the control.

The experimental field was treated with conventional moderate deep ploughing by a YMM 20 tractor after harvesting potatoes from the previous season. Rabbit and poultry manure fertilizer was then added, and the lentils were sown manually in lines with 25 mm sowing depth, 25 mm seeding distance and 20 mm inter-row spacing. The plots were finally pressed by roller [11].

The ZnO-NP solution was mixed with adjuvant SILWET STAR^®^ (Chemtura Manufacturing UK Limited, Manchester, UK) to final 1 mg·L^−1^ NP concentration before direct foliar application. This eased nanoparticle penetration through the plant leaf wax sub-structure, and the adjuvant was applied without ZnO-NPs in the control experiment. The ZnO-NPs solution was applied twice during the growth season as recommended by Meier [38]. This occurred on the 45th and 60th day of the vegetation period when lentils achieved their ideal phenological growth stage (Figure 2). The solution was dispersed on the plant on a windless early morning by GAMMA5 pressure sprayer until the leaves were completely wet (GAMMA 5, Mythos Di Martino, Mussolente, Italy). The standard treatment included weed removal twice during the growing season by hand weeder. One liter of spray liquid was applied for each experiment replication.

### 2.5. Analysis of Quantitative Parameters; Water Stress, Stomatal Conductance, Plant Temperature and Temperature Difference

The lentil plants in each treatment were harvested manually when seeds attained physiological maturity. Approximately 10% of the plants began to change to yellow color at that time, and the lower-level pods turned to yellow-brown and brown [39]. The number of plants and number of pods per plants were calculated manually. The plant height was determined in millimeters by Texi 4007 laboratory equipment (Texi GmbH, Berlin, Germany). The seed yield was then determined in grams on the KERN PC3500-2 laboratory scale (Kern & Sohn, Balingen, Germany), and the thousand-seed weight was analyzed by NUMIREX equipment (MEZOS, Hradec Králové, Czech Republic).

Non-intrusive measurement of plant temperature (°C), temperature difference (°C), crop water stress index (CWSI) and stomatal conductance index (Ig) were performed from 11:00 a.m. to 1:00 p.m. periodically from the 46th to 100th day of the growth season. Ten measurements were made diagonally on each leaf of ten similar plants in each replication, as in Jones et al. [40].

The temperature difference is calculated from atmospheric and plant temperatures (Equation (1)). The plant is under stress when the temperature difference has zero or positive value, and it has greater environmental stress resistance when the value is negative. The lower plant temperatures are due to intensive transpiration.
*T**^dif^* = *T**^leaf^* − *T**^air^*(1)

The CWSI calculation was performed as in Jones et al. [40] using EasIR-4 thermo-camera (Bibus AG, Fehraltorf, Switzerland). The measurements included leaf temperature parameters (*T^leaf^*), dry (*T^dry^*) and wet (*T^wet^*) leaf surface and atmospheric moisture (Equation (2)).
(2)$$CWSI=\frac{\left(T^{leaf}-T^{wet}\right)}{\left(T^{dry}-T^{wet}\right)}\;$$

The thermal lentil measurements were recorded from 2 m distance and 150 cm height, with the auto-focused field at 20.6 × 15.5 [12]. This provided the stomatal conductance index calculation in Equation (3).
(3)$$Ig=\frac{\left(T^{dry}-T^{leaf}\right)}{\left(T^{leaf}-T^{wet}\right)}\;$$

### 2.6. Seed Mineral Composition Analysis

The final lentil seed mineral concentration was analyzed after digestion of 0.15–0.30 g seed samples. This was performed by the Anton Paar Multiwave 3000 microwave digestion system (Graz, Austria) in PTFE pressure vessels using the concentrated HNO_3_ and H_2_O_2_ mixture at 60 bar pressure. Iron, zinc, magnesium, calcium and phosphor content was determined by ICP-OES (Varian Vista MPX, Mulgrave, Victoria, Australia) with yttrium as the internal standard. Potassium, copper and manganese were then provided by F-AAS (Perkin-Elmer Model 1100, Waltham, MA, USA) [41]. Finally, the Kjeldahl method determined total nitrogen, and colorimetry established sulfur content.

### 2.7. Soil X-ray Diffraction Analysis and Verification of Zinc Oxide Nanoparticles

X-ray powder diffraction analyzed (XRD) the soil mineral composition. This was accomplished by Philips, Netherlands PW 1710 diffractometer with Cu-anode and graphite monochromator at 40 kV voltage and 20 mA electric current.

X-ray diffraction by a Bruker D8 DISCOVER diffractometer (Bruker, MA, USA) verified zinc oxide nanoparticles. The measurement conditions were 12 kW, 40 kV and 300 mA with a Cu-anode similar to that in Kolenčík et al. [11].

### 2.8. Scanning Electron Microscopy of ZnO-NPs

The NP-suspension was diluted and transferred to a copper mesh with a carbon/formvar membrane. The ZnO-NP grid was then placed in a holder and microscope chamber after drying. The ZnO-NPs morphology and size were observed in transmission mode under the JEOL JSM 7610F+ scanning electron microscope with Schottky cathode (SEM—JEOL, Tokyo, Japan). Dark- and bright-field observation at 30 keV accelerating voltage provided comparison.

### 2.9. Statistical Analysis

Statistical analysis was performed by STATISTICA software (StatSoft, Tulsa, OK, USA). The normality of experimental data was then assessed by Shapiro–Wilks test, and statistically significant differences between treatments were achieved using the Student’s two-sample *t*-test and ANOVA followed by Fisher’s least-significant difference.

## 3. Results and Discussion

### 3.1. Colloidal Properties of ZnO-NPs Foliar Application

The applied ZnO-NP physical–chemical properties have previously been reported [11]. Herein, we determined the wurtzite-type ZnO-NP crystal structure (Figure 3b). The mean diameter of individual particles was 17.3 ± 0.1 nm [11] with spherical, hexagonal, columnar, rod-like and cuboidal shape (Figure 3a).

The detected particle size has a profound effect on ZnO-NP availability. Although NPs usually enter plants through the stomata or cuticle, cuticular access is size-restricted to particles with diameter under 5 nm, and therefore ZnO-NPs have only stomatal transport available [42]. Transport velocity is also partly limited. However, ZnO-NP exposure to sunlight can initiate photo-corrosion, which is usually associated with transformation to more bioavailable Zn^2+^ species. These ions are complexed to organic acids and transported to plant tissues through cuticle access [43].

The individual NP size range is less than the 282.5 nm hydrodynamic diameter determined in the solution prepared for foliar application (Table 2).

Hu et al. [44] recorded that smaller inorganic NPs with higher zeta-potential can penetrate the leaf’s lipid bilayers more efficiently for greater effect on the plant. Bhattacharjee [45] also reported that zeta-potentials below −30 mV indicate NP colloidal instability in solution. Therefore, our determined zeta-potential of −33.3 mV in Table 2 should prevent ZnO-NP aggregation and sedimentation.

### 3.2. ZnO-NP Effect on Lentil Quantitative Parameters

Zinc is required by plants for carbohydrate metabolism and gene expression in response to environmental stress [46], and ZnO-NPs are applied as nano-fertilizers and agrochemicals to enhance crop production [20,47,48]. Liu and Lal [49] recorded that the advantage of nano-domains is their gradual release of beneficial nutrients, and this encourages the most effective plant growth and development. Photocatalytic effects similar to those reported in TiO_2_-NP use induce photosynthesis during that slow release [50,51]. Singh et al. [52] noted that ZnO-NPs’ positive effect on plant growth is achieved in laboratory conditions at NP concentrations as high as 60 mg·L^−1^, and Prasad et al. [47] proposed even higher effective concentrations.

Statistically significant differences for quantitative parameters were observed when 1 mg·L^−1^ suspension of ZnO-NPs was applied. These included the number of pods per plant, seed yield and thousand-seed weight. This contrasted with our NP-free control, and there was also no statistical difference observed in the number of plants or their height in the NP-suspension experiment (Table 3). Our results are supported by the foliar application of 20 mg·L^−1^ ZnO-NPs to *Abelmoschus esculentus* which increased its pod number and improved plant growth and yield [53].

### 3.3. Soil Nutrient Source, Bioavailability and Nutritional Quality of Harvested Lentil Seeds

The quality of lentil seeds can be determined by various parameters [54], including the essential and beneficial nutrient content [8,55]. Unfortunately, there is little knowledge of the nano-material effect on mineral nutrient uptake and overall crop nutritional quality [55].

The experimental area has not been exposed to widespread or long-term application of chemical fertilizers. Therefore, soil release of macro, micro and trace nutrients to lentil plants is partly governed by their spontaneous release from the naturally occurring soil minerals (Figure 4). The dominant soil minerals determined at the experimental field were quartz (SiO_2_), muscovite [KAl_2_(Si_3_Al)O_10_(OH,F)_2_], calcite (CaCO_3_), dolomite (MgCO_3_), and chlorite [(X,Y)_4−6_(Si,Al)_4_O_10_(OH,O)_8_, where the position of “X” and “Y” correspond to elements such as Fe^2+^, Fe^3+^, Ni^2+^, Zn^2+^, Al^3+^, Li^+^, or Ti^+4^], K-feldspar (KAlSi_3_O_8_), Na-feldspar (NaAlSi_3_O_8_) and Ca-feldspar (CaAl_2_Si_2_O_8_).

Table 4 lists the plant micronutrients available from the extractable fraction of the experiment soil. All determined bioavailable Cu, Fe and Mn values correspond to results reported by Bloem et al. [56]. Zinc, however, is one exception because its higher content most likely originated from residues deposited in the soil after ZnO-NP foliar application.

Several studies have evaluated NP effects on lentil production. These include biosynthesized Au-NPs [57], SiO_2_-NPs [58], TiO_2_-NPs [59], ZnO-NPs [60] and Ag-NPs [61]. The studies, however, do not consider the changes in content of nutrients such as minerals which affect final lentil seed quality. Our foliar application of low ZnO-NPs concentration did not significantly affect the lentil concentration of some essential and beneficial nutrients, including Zn (Table 5). The Zn concentration accumulated in the seeds is not statistically different in the ZnO-NP-treated plants and the control. In addition, the Zn concentration indicated in Table 5 is similar to that noted by Grusak [62]. Although it appears that foliar ZnO-NP application did not affect lentil seed Zn content, it may have influenced other physiological parameters such as photosynthesis [63], because zinc can be a fundamental component in photosynthetic processes [13]. Erdal et al. [64] noted that Zn fertilization led to decreased phosphor content, and it was further reported that the ZnO-NPs interrupted iron uptake and encouraged nitrogen assimilation [55]. In contrast, the nitrogen concentration in both variants corresponded to the typical nitrogen content in lentil seeds [8]. This was despite our results indicating that the nitrogen content in lentil seeds was lower when the plants were treated with ZnO-NPs. This ZnO-NP exposure resulted in decreased iron content in the lentil seeds, and although lentil is usually high in iron [65], both treated and control plants had lentil seed concentrations significantly below the typical average value [8]. This was most likely due to internal factors in our experiment plant variety [29]. Finally, there were no statistically significant differences in phosphor content as a seed anti-nutrient in ZnO-NP-treated plants and the control. The values corresponded with the reported average [62].

A similar trend was observed in potassium, calcium, sulfur, manganese, copper and magnesium. The seed contents did not deviate from expected values [8,62], and was unaffected by ZnO-NP treatment. There were no statistically significant differences in the analyzed seed nutrient contents of treated and untreated lentil plants, but all average content values except potassium were slightly higher in the control. This may have been due to improved stress response or dilution effect because the ZnO-NP biomass yield was higher. This effect has been noted by other authors [11,66,67].

### 3.4. Stress Response to ZnO-NPs Foliar Application of Lentil

We recorded statistically significant differences in stomatal conductance and transpiration efficiency in ZnO-NP-treated plants and the control (Table 3). Stomatal conductance was enhanced despite the applied low ZnO-NP concentration. Although Faizan et al. [63] applied a 50-fold higher ZnO-NP concentration, they recorded a similar response. Statistically significant difference in crop water stress index (CWSI) was also noted between the ZnO-NP-treated plants and the control (Table 3).

The higher control CWSI indicated that ZnO-NP application eased the general effect of environmental stress on the lentil plant, including its response to water-deficiency and higher air-temperatures [68,69]. Statistically significant ZnO-NP difference was also confirmed for plant temperature and temperature difference compared to the untreated controls (Table 3). Moreover, all physiological indices show that ZnO-NP foliar application affected photosynthesis. This is supported by the increased leaf chlorophyll content in wheat [70] and okra [71].

There were no statistically significant differences between ZnO-NP-treated plants and the controls during leaf development or prior to the first ZnO-NP application. However, statistically significant differences were detected in the selected physiological indicators following NP exposure and in the following lentil growth stages (Figure 5a–d).

The positive lentil response to NPs was apparent for all physiological indices within three weeks of ZnO-NP treatment. A similar response was confirmed in our previous study, where the common sunflower was exposed to low ZnO-NP foliar concentration [12]. The improved stress response was maintained after the second foliar application, and this highlighted increased lentil yield and thousand-seed weight (Table 3).

The stress incurred was due to altered environmental conditions, and NP treatment resulted in increased plant resistance. In addition, Salama et al. [72]. noted increased production and quality in this dry bean following exposure to 10–40 mg·L^−1^ ZnO-NPs.

Figure 5 shows the seasonal dynamics, and our results of the average daily air temperature and calculated CWSI, Ig, T_p_ and T_d_ indices concur. Figure 5 statistically significant difference of *p* = 0.001 was recorded for all physiological indices when the average air temperature was above 19 °C (Figure 6a) and plants were exposed to 10 h minimum sunshine (Figure 6b). Lentil is sensitive to photo-thermal conditions and its response to temperature and photoperiod is expressly linear [6]. It is, therefore, highly likely that the improved stress response to NPs is enhanced when the photo-thermal conditions overcome the critical 10 h photoperiodic lighting and corresponding temperature, and the lentil seed yield is then higher [6].

Rainfall precipitation also appears partly responsible for the observed effects. The statistically significant differences for almost all measured physiological indices were primarily noted when the average weekly precipitation was less than 18 mm (Figure 6c). This is important because weekly precipitation above 19 mm decreases average atmospheric temperature. Higher precipitation with subsequent slightly higher soil water content was confirmed on the 87th day when increased lentil transpiration was an observed trend. One stomatal conductance index was the only statistically significant difference at that time.

## 4. Conclusions

Our field experiment applied low 1 mg·L^−1^ ZnO-NP foliar spray dispersion to the Lens culinaris ´Beluga´ lentil. It provided positive ZnO-NP production responses, including increased lentil seed yield, thousand-seed weight and the number of pods per plant. This strongly contrasted with the untreated control. However, the Zn foliar supplementation did not affect its translocation to seeds regardless of Zn content, and therefore did not change the lentil seed nutrient quality.

Although there were no statistically significant differences in treated and untreated lentil seed nutrients, slightly higher average values were noted in the control, except for potassium. The ZnO-NP-treated plants had improved stress response within three weeks of NP exposure, and this included differences in stomatal conductance, crop water stress index, plant temperature and temperature difference.

In conclusion, our results justify the increasing trend of nano-fertilizer application in sustainable agriculture. The application of low NP concentrations has a significant impact on plant stress responses. This enhanced lentil seed quantitative properties. Finally, these procedures highlight the pro-environment character of the agricultural practices in our study, and the low ZnO-NP concentrations herein should not over-burden the increasing environmental nanoparticle input.

## Figures and Tables

**Figure 1 nanomaterials-12-00310-f001:**
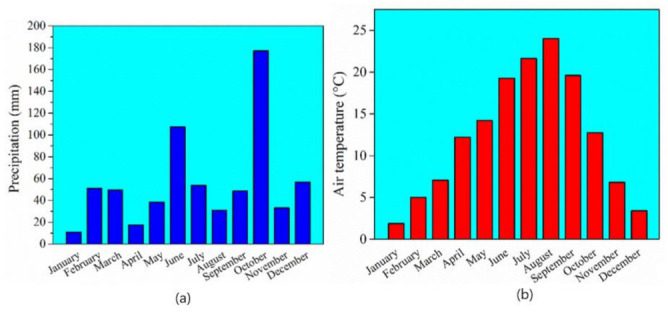
Monthly variations in (**a**) precipitation, (**b**) air temperature and (**c**) hours of sunshine during the 2020 vegetation season at the Hostie Village near Zlaté Moravce in Slovakia.

**Figure 2 nanomaterials-12-00310-f002:**
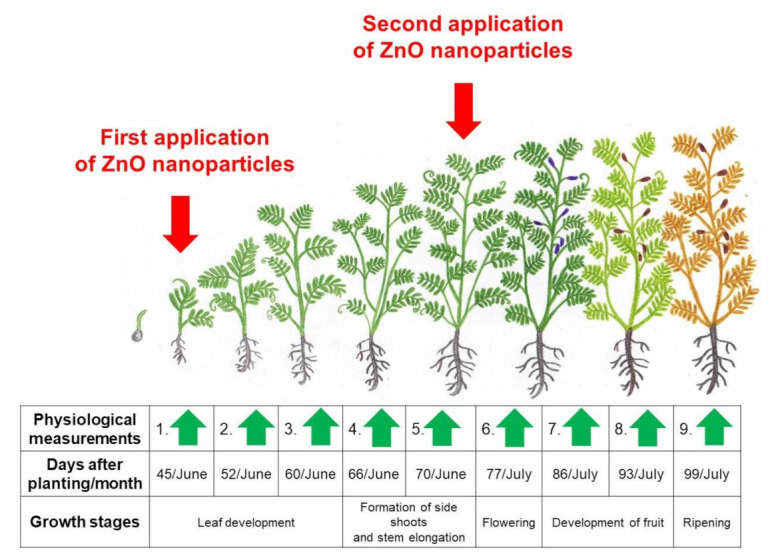
Schematic model of lentil growth stages. The red arrow shows the foliar application of 1 mg·L^−1^ ZnO-NPs, and the green arrows indicate the physiological measured time-intervals to assess lentil development.

**Figure 3 nanomaterials-12-00310-f003:**
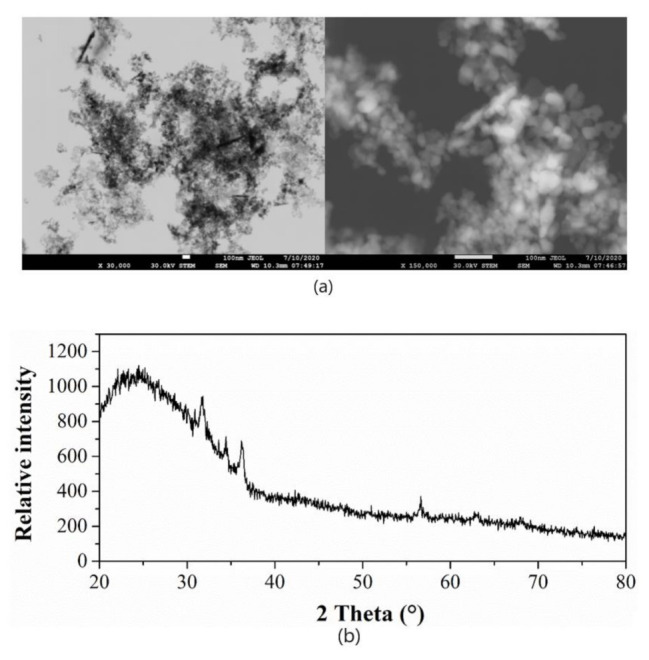
(**a**) Scanning transmission electron microscopy (STEM) visualized the zinc oxide nanoparticles (ZnO-NPs) used for foliar application to the lentils. The left STEM micrograph is bright-field, and dark-field is on the right, and (**b**) X-ray diffraction analysis shows the zinc oxide nanoparticles have wurtzite-structural symmetry.

**Figure 4 nanomaterials-12-00310-f004:**
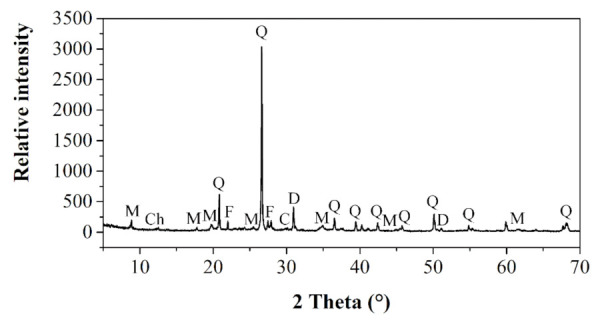
X-ray diffraction powder patterns of soil collected at the Hostie experimental locality at Zlaté Moravce in Slovakia; dominant quartz (Q), muscovite (M), chlorite (Ch), calcite (C), dolomite (D) and feldspar (F) minerals.

**Figure 5 nanomaterials-12-00310-f005:**
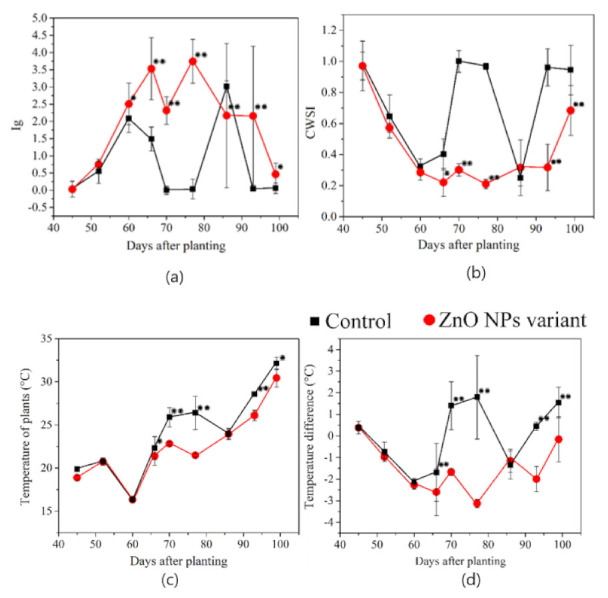
Seasonal effects of ZnO-NP foliar application on (**a**) stomatal conductance index (Ig), (**b**) crop water stress index (CWSI), (**c**) plant temperature (Tp) and (**d**) temperature difference (Td) compared to untreated controls (significance: * *p*-value < 0.05, ** *p*-value < 0.001).

**Figure 6 nanomaterials-12-00310-f006:**
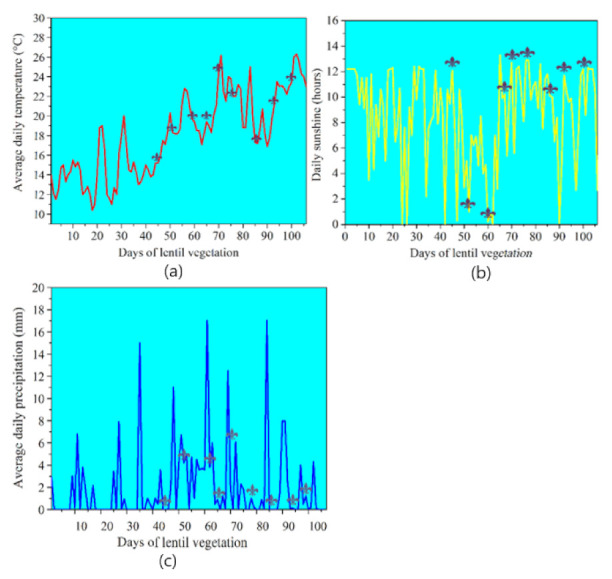
Average (**a**) daily air temperature, (**b**) daily sunshine, and (**c**) daily precipitation during the entire 106-day lentil vegetation season at the Hostie experimental locality at Zlaté Moravce in Slovakia. This period covered sowing on 20 April 2020 to harvesting on 3 August 2020. The inserted grey markers indicate the measurement day.

**Table 1 nanomaterials-12-00310-t001:** Selected soil features from the Hostie experimental locality at Zlaté Moravce in Slovakia prior to experimental treatments.

pH_H2O_	pH_KCl_	^1^CC (%)	^2^HA (mmol·kg^−1^)	^3^SEBC (mmol·kg^−1^)	^4^TSC (mmol·kg^−1^)	^5^DSCBC (%)	^6^EC (S)	^7^C _org_ (%)
7.1 ± 0.1	6.9 ± 0.02	4.6 ± 1.0	5.0 ± 0.3	465 ± 13	470 ± 13	98.9 ± 0.1	310 ± 46	2.5 ± 0.5

^1^CC—carbonates’ content; ^2^HA—hydrolytic acidity; ^3^SEBC—sum of exchangeable basic cations; ^4^TSC—total sorption capacity; ^5^DSCBC—degree of saturation of complex with basic cations; ^6^EC—electrical conductivity, ^7^C _org_—soil organic carbon

**Table 2 nanomaterials-12-00310-t002:** ZnO-NP colloidal properties in foliar application.

Hydrodynamic Diameter (nm)	Zeta-Potential (mV)	Electrophoretic Mobility (cm^2^·V^−1^·s^−1^)	Conductivity (mS·cm^−1^)
282.5 ± 6.9	−33.3 ± 0.8	−25.7·10^−5^ ± 0.8·10^−5^	0.070 ± 0.002

**Table 3 nanomaterials-12-00310-t003:** Comparison of the quantitative and physiological parameters of ZnO-NP-treated lentils and the NP-free control recorded in the 2020 vegetation season. The values state the means and standard deviation, and these were compared by Fisher’s least-significant difference (significance: * *p*-value < 0.05; ** *p*-value < 0.001.).

	ZnO-NPs Foliar Applied Variant	Control Variant (Without ZnO-NPs Application)
*Quantitative parameters*		
Number of plants (pcs)	110 ± 3	109.3 ± 5
Height of plants (mm)	515 ± 42	535 ± 35
Number of pods per plant (pcs)	49 ± 5 *	35 ± 6 *
Weight of thousand seeds	17.4 ± 1.0 *	14.2 ± 1.0 *
Seed Yield (g)	35.5 ± 5.1 *	21.5 ± 4.3 *
*Physiological parameters*		
Temperature of plants (°C)	22.57 ± 0.4 **	24.04 ± 0.2 **
Temperature difference (°C)	−1.51 ± 0.05 **	−0.04 ± 0.03 **
Stomatal conductance index (Ig)	1.9 ± 0.1 **	0.8 ± 0.1 **
Crop water stress index (CWSI)	0.4 ± 0.1 *	0.7 ± 0.2 *

**Table 4 nanomaterials-12-00310-t004:** Content of soil-extractable micronutrients expressed as mg·kg^−1^ soil.

Elements	Extractable Micronutrients (mg·kg^−1^ Soil)
Zinc	15 ± 1.1
Manganese	27.5 ± 1.4
Copper	2.28 ± 0.1
Iron	24.7 ± 1.1

**Table 5 nanomaterials-12-00310-t005:** ZnO-NP foliar application effects on g·kg^−1^ content of selected essential and beneficial nutrients in the harvested lentil seeds.

	ZnO-NPs-Treated Plants	Plants That Were Not Exposed to ZnO-NPs
Zinc	0.05 ± 0.003	0.05 ± 0.03
Phosphor	5.8 ± 0.3	6.1 ± 0.3
Nitrogen	42.7 ± 2.1	47.1 ± 2.4
Potassium	11.2 ± 0.6	10.8 ± 0.5
Sulfur	3.13 ± 0.32	3.38 ± 0.29
Copper	0.013 ± 0.005	0.013 ± 0.005
Iron	0.05 ± 0.002	0.06 ± 0.001
Manganese	0.009 ± 0.001	0.012 ± 0.002
Calcium	0.52 ± 0.03	0.58 ± 0.03
Magnesium	1.01 ± 0.05	1.04 ± 0.05

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
