# Peer review of "Effects of Foliar Application of ZnO Nanoparticles on Lentil Production, Stress Level and Nutritional Seed Quality under Field Conditions"

_nanomaterials, 2022, doi:10.3390/nano12030310_

Round 1
Reviewer 1 Report
The manuscript of Marek Kolenčík and co-authors is devoted to the study of the effect of ZnO NPs on the growth of lentil under field conditions. In my opinion, the work has not received a sufficient amount of significant data for publication in this journal.
Main comments
1) The novelty of this study is not clear. The authors themselves point out that several publications, including two previous publications by the authors [9, 10], have shown a positive effect of ZnO NPs on plant growth. What is fundamentally new is shown in this article, in addition to another object of research – lentil?
2) The article contains a section “characterization of nanoparticles”. However, the authors have used Sigma-Aldrich ZnO nanoparticles (721077) here and in their two previous articles [9,10]. In particular, the information “a diameter size of individual particles was 17.3±0.1 nm” is also given in [9]. What is the significance of the data given in the Tabl. 2?
3) In my opinion, in the article the influence of ZnO nanoparticles on the physiological parameters of lentil is insufficiently studied. The data obtained by the authors on the change of stomatal conductance index and crop water stress index do not provide a clear picture of how exactly the ZnO nanoparticles affect the physiology of lentil plants.
4) In my opinion, plant growth in the field is influenced by many factors, and not all of them can be taken into account. Therefore, such data should be compared with data obtained for plant growth under more standardized laboratory conditions.
Additional comments
Abstract
- The abstract is written in general terms and does not give an idea of what was done and what results were obtained.
- What do the authors mean by “anti-nutrients” here and in the conclusion?
Introduction
It is worth adding information about the toxicity of nanoparticles and harm to the environment, in the conclusion the authors mention this.
Methods
- 2.1. Which variety of lentil was used?
- 2.2. The description of nanoparticles is given in [9].
- Tabl. 1. What was the impact of these data? What they are for here?
- Tabl. 2. Is this the average temperature for the month? Deviations are not indicated.
- 2.4 What is this adjuvant? What is its effect on plants?
- 2.5. The authors gave the concentration of the solution of ZnO NPs. At the same time, it is not clear what their dose was per plant, per kg of biomass, etc.
- 2.6. Worth using seeds instead of fruit
- 2.7. Why the soil was analyzed and at what point in the experiment. What is the significance of the data obtained?
Results and Discussion
These sections should be listed separately. Now the article goes to point 3, and then 5.
- 3.1. lines 181-182 “The detected particle size has an imminent effect on the availability of ZnO-NPs.” This is not investigated in the article, only literature data are given.
- Tabl. 3. Yield (g) - how many plants is it? In Fig. 4, the authors show the fluctuations of the stomatal conductance index and crop water stress index from time to time. How then the data in Tabl. 3 are obtained?
- 3.3. Why this section “Nutrient quality of harvested lentil seeds” provides soil analysis?
- Tabl. 4. No significant difference was found. But the authors nevertheless discuss these results in detail and they quote in the conclusion section the phrase “However, the foliar supplementation of Zn did not affect its translocation into seeds and, thus, did not change the nutrient quality of lentil seeds regarding the Zn content. However, the content of some essential and beneficial anti-nutrients was slightly altered.”
- 3.4. lines 265-266 “Both physiological indexes suggested that foliar application of ZnO NPs affected the photosynthesis”. This was not investigated in the article. Perhaps this was due to the mechanical effect of nanoparticles of this size. What is known about similar effects of other nanoparticles?
- lines 275-276 What does Figure 3 have to do with it?
- Fig. 4a,b contains outliers at day 87 after planting. The authors do not write about it. What is the reason for outliers?
- Fig. 5 and lines 287-298 There are many environmental factors under field conditions that are difficult to take into account. The effects of different temperatures and amounts of moisture can be assessed when plants are grown in laboratory conditions at their constant values.
Author Response
Reviewer 1
Extensive editing of English language and style is required.
Answer: Manuscript has been edited by the native English-speaking lector Dr. Raymond Joseph Marshall.
Comment #1
The novelty of this study is not clear. The authors themselves point out that several publications, including two previous publications by the authors [9, 10], have shown a positive effect of ZnO NPs on plant growth. What is fundamentally new is shown in this article, in addition to another object of research – lentil?
Answer: The novelty and adding value of our research have been highlighted in various parts of the manuscript e.g., in abstract section “…foliar application of low ZnO-NP concentrations had promising results for increasing crop production under field conditions, and this confirms ZnO-NP use as a viable strategy for sustainable agriculture…”, in introduction “Lack of research into these aspects inspired us to use foliar ZnO-NP administration to lentils under field conditions to provide new information on ZnO-NPs’ effects on these plants.”, in results and discussion “Soil nutrient source, bioavailability and nutritional quality of harvested lentil seeds”, and in concluding remarks as well “We established positive production responses to ZnO-NPs, including increased lentil seed yield, thousand-seed weight and the number of pods per plant.”.
Comment #2 & Comment # 7.2
The article contains a section “characterization of nanoparticles”. However, the authors have used Sigma-Aldrich ZnO nanoparticles (721077) here and in their two previous articles [9,10]. In particular, the information “a diameter size of individual particles was 17.3±0.1 nm” is also given in [9]. What is the significance of the data given in the Tabl. 2?
Answer: To resolve this issue, the name of sub-chapter has been changed to “Colloidal properties of the foliarly applied ZnO-NPs”. and the following information was highlighted… “Hu, et al. [41] concluded that the smaller inorganic NPs with higher zeta potential can translocate more efficiently through the leaf’s lipid bilayers and have more prominent impact on the plant....”, which indicates the necessity of evaluating the colloidal properties of NPs in actual solution that was applied to the plants that can be slightly altered after dilution of commercially available NPs.
Comment #3
In my opinion, in the article the influence of ZnO nanoparticles on the physiological parameters of lentil is insufficiently studied. The data obtained by the authors on the change of stomatal conductance index and crop water stress index do not provide a clear picture of how exactly the ZnO nanoparticles affect the physiology of lentil plants.
Answer: To address this issue, the term “physiology” has been completely reevaluated for the most of parts of the manuscript since we have realized that it was too general. Additionally, we have introduced another physiological indices into the manuscript to upgrade the quality of our results and to point out the importance of “physiological” stress responses.
Comment #4
In my opinion, plant growth in the field is influenced by many factors, and not all of them can be taken into account. Therefore, such data should be compared with data obtained for plant growth under more standardized laboratory conditions.
Answer: We appreciate the suggestion from the reviewer, however, the current manuscript studies the field experiment that is characteristic for the unpredictable weather conditions. Thus, the effect of ZnO NPs on lentil in real soil environment is the focus here. It is also one of the adding values (see our response to comment #1). On the other side, we have considered this comparison to the lab experiment under controlled conditions, however, this type of experiment does not fully correspond to real field experiments.
Comment #5
- The abstract is written in general terms and does not give an idea of what was done and what results were obtained.
Answer: As suggested, the abstract section has been completely rewritten to more logical framework, and there have been added new information according to reviewer’s suggestions.
Comment #5.1 What do the authors mean by “anti-nutrients” here and in the conclusion?
Answer: This term has been introduced to distinguish the physiological and nutritional properties of some nutrients in lentils on the human organism. The significance of these different properties have been introduced as follows: “This balance in fully ripe lentil seeds is important for higher phytic acid content because this absorbs further micronutrients and contains phosphor as an antinutrient. Intestinal digestion further negatively affects nutrient bioavailability [27]”.
Comment #6 Introduction
It is worth adding information about the toxicity of nanoparticles and harm to the environment, in the conclusion the authors mention this.
Answer: Since the toxicity of nanoparticles was not the intention of our research (application of agrochemicals to enhance growth parameters), we did not consider their negative effect in manuscript and thought of this information as redundant.
Comment #7 Methods
Comment #7.1 Which variety of lentil was used?
Answer: We have used black lentil (Lens culinaris) ´Beluga´.
Comment #7.2 The description of nanoparticles is given in [9].
Answer: Please, check our answer to Comment #2.
Comment #7.3 & Comment 7.8 & Comment 8.3 & Table. 1. What was the impact of these data? What they are for here? Comment #7.4 Table. 2. Is this the average temperature for the month? Deviations are not indicated.
Answer: In the case of Table 1 & Figure 1 (a-c), the determined soil properties and monthly environmental conditions per year are usually introduced without SD. (Reviewer most likely means Figure 1 because the aforementioned Table 2 only indicates the colloidal properties of NPs). Table 1 & Figure 1 were introduced specifically for the agronomically orientated readers who incline to compare the soil and surrounded environmental conditions in our study to other real agronomic localities. To even further increase the agronomical value of the data, the new figure indicating the time of exposure to sunshine has been added to the revised manuscript. Similar useful data for agronomical practice were also presented by Liu et al. (2022). Additionally, for evaluation of soil bioavailable metals, the extractable data on their concentrations were added to manuscript in form of Table. 4, and it was introduced in the is manuscript as follows, “Table 4 lists plant micronutrients available from the extractable fraction of the experiment soil. All determined bioavailable Cu, Fe and Mn values correspond to results reported by Bloem, et al. [51]. Zinc is one exception because its higher content most likely originated from residues deposited in the soil after ZnO-NPs foliar application.”.
Comment #7.5 What is this adjuvant? What is its effect on plants?
Answer: To address this issue, the following information has been added to the manuscript …” The ZnO-NP solution was mixed with adjuvant SILWET STAR® to final 1mg×L-1 NPs concentration before direct foliar application. This eased NPs penetration through the plant leaf wax sub-structure, and the adjuvant was applied without ZnO-NPs in the control experiment.
Comment #7.6 The authors gave the concentration of the solution of ZnO NPs. At the same time, it is not clear what their dose was per plant, per kg of biomass, etc.
Answer: This information has been a part of the manuscript as follows: “The solution was dispersed on the plant by GAMMA5 pressure sprayer during a windless early morning until the leaves were completely wet. (GAMMA 5 - Mythos Di Martino, Mussolente, Italy).”.
Comment #7.7 Worth using seeds instead of fruit
Answer: We agree, it was change according to reviewer’s suggestion.
Comment #7.8. Why the soil was analyzed and at what point in the experiment. What is the significance of the data obtained?
Answer: Please, check our response to Comment #7.3.
Comment #8. Results and Discussion
These sections should be listed separately. Now the article goes to point 3, and then 5.
Answer: Since it is applicable in the journal to present and discuss data in one section, we decided to do not split our manuscript into the aforementioned sections to maintain the consistency of the manuscript.
Comment #8.1 lines 181-182 “The detected particle size has an imminent effect on the availability of ZnO-NPs.” This is not investigated in the article, only literature data are given.
Answer: We agree, thus, this paragraph has been rewritten.
Comment #8.2 Tabl. 3. Yield (g) - how many plants is it? In Fig. 4, the authors show the fluctuations of the stomatal conductance index and crop water stress index from time to time. How then the data in Tabl. 3 are obtained?
Answer: This information has been added to manuscript as follows: “The lentil plants in each treatment were harvested manually when seeds attained physiological maturity. Approximately 10% of the plants began to change to yellow color at that time, and the lower-level pods turned to yellow-brown and brown [37]. The number of plants and number of pods per plants were calculated manually. The plant height was determined in millimeters by Texi 4007 laboratory equipment (Texi GmbH, Berlin, Germany). The seed yield was then determined in grams by the KERN PC3500-2 laboratory scale (Kern & Sohn, Balingen, Germany), and the thousand-seed weight was analyzed by NUMIREX equipment (MEZOS, Hradec Králové, Czech Republic)...”.
Comment #8.3 Why this section “Nutrient quality of harvested lentil seeds” provides soil analysis?
Answer: Please, check our response to Comment #7.3.
Comment #8.4 Tabl. 4. No significant difference was found. But the authors nevertheless discuss these results in detail and they quote in the conclusion section the phrase “However, the foliar supplementation of Zn did not affect its translocation into seeds and, thus, did not change the nutrient quality of lentil seeds regarding the Zn content. However, the content of some essential and beneficial anti-nutrients was slightly altered.”
Answer: To address this issue, the aforementioned section was rewrite according to reviewer’s suggestion “While there were no statistically significant differences in the analyzed lentil seed nutrients for both variants, slightly higher average values were recorded for the control treatment.”.
Comment #8.5 lines 265-266 “Both physiological indexes suggested that foliar application of ZnO NPs affected the photosynthesis”. This was not investigated in the article. Perhaps this was due to the mechanical effect of nanoparticles of this size. What is known about similar effects of other nanoparticles?
Answer: We agree, therefore, this paragraph has been rewritten and new physiological indicators have been added support our conclusions, including the data on Temperature of plants and Temperature difference”. Also new references have been introduced to provide a supportive information to our claims “…development. That release also induces photosynthesis, based on photocatalytic effect similar to TiO2-NPs [46,47].”
Comment #8.5 lines 275-276 What does Figure 3 have to do with it?
Answer: We agree, thus, the new subchapter “Soil nutrient source, bioavailability and nutritional quality of harvested lentil seeds” has been introduced in the manuscript.
Comment #8.6 Fig. 4a,b contains outliers at day 87 after planting. The authors do not write about it. What is the reason for outliers?
Answer: To address this issue, the following information has been added: “This is important because average weekly precipitation above 19mm decreases the atmospheric temperature, and this higher precipitation associated with slightly higher soil water content was confirmed by measurements on the 87th day. There was increased lentil transpiration, and no statistically significant differences were noted for any physiological indices at that time.”.
Comment #8.7 Fig. 5 and lines 287-298 There are many environmental factors under field conditions that are difficult to take into account. The effects of different temperatures and amounts of moisture can be assessed when plants are grown in laboratory conditions at their constant values.
Answer: We appreciate your suggestion to carry out the lab experiments, however, as already mentioned in our previous responses, the lab conditions will never correspond to effects of unpredictability of real environment what is the aim and the novelty of our study as well. Additionally, for make a stress response clearer, we have introduced “seasonal” graph of daily hours of sunshine during lentil vegetation as follows: “Further, the p = 0.0001 statistically significant difference was noted for all physiological indices (Figure 4) when the average air temperature was above 19°C (Figure 5a) and the plants were exposed to 10 hours minimum sunshine (Figure 5b). Lentil is sensitive to photo-thermal…”.
Reviewer 2 Report
The manuscript described the effect of ZnO nanoparticles (NPs) on physiological parameters, yield, and seed quality of lentils. The authors claimed that foliar application ZnO NPs significantly increased some plant physiological parameters, yield, and quality of lentil seeds but decreased contents of various macro and micronutrient elements including Zn in the seeds compared to untreated control. Although the topic is generally interesting, however, the study plan, presentation of data, and discussion are very weak. I have some major comments as follows:
- The title may be revised as "Effect of foliar application of ZnO nanoparticles on physiology, yield and seed quality of lentil".
- Abstract: The first sentence should be replaced by a general statement on the problem of the study by replacing the current sentence. The Abstract must include the summary results and perspective of the findings.
- The Zn is an essential nutrient element. Application of ZnO NPs added Zn to the plants. Why equivalent amount of Zn was not applied to the control plants by Zn fertilizer?
- Results and discussion section needs substantial revision or rewriting. First, the results of the experiment must be presented/described. Second, discuss the novelty and perspective of the findings with relevant literature. This manuscript did not follow this sequence. For example, 3.2 subsection is written like a review article!
- Table 3 should be revised with units of the obtained data.
- What is phosphor in Table 4? Why do all the elements in the lentil seeds decrease?
- Figure 5 should not be included in the results section. ZnO NPs treated plants although the yield and physiology were improved.
Author Response
Reviewer 2
Extensive editing of English language and style is required.
Answer: Manuscript has been edited by the native English-speaking lector Dr. Raymond Joseph Marshall.
Comment #1 The title may be revised as "Effect of foliar application of ZnO nanoparticles on physiology, yield and seed quality of lentil".
Answer: The title of manuscript was changed according to reviewerʼs suggestion as follows: “Effects of foliar application of ZnO nanoparticles on lentil production, stress level and nutritional seed quality under field conditions”.
Comment #2 Abstract: The first sentence should be replaced by a general statement on the problem of the study by replacing the current sentence. The Abstract must include the summary results and perspective of the findings.
Answer: As suggested, the general statement, summary and future perspectives of our findings were added to the abstract as follows: “Nanotechnology offers new opportunities for the development of novel strategies and materials which improve technology and industry sectors.”, and “Therefore, foliar application of low ZnO-NP concentrations had promising results for increasing crop production under field conditions, and this confirms ZnO-NP use as a viable strategy for sustainable agriculture.”.
Comment #3 The Zn is an essential nutrient element. Application of ZnO NPs added Zn to the plants. Why equivalent amount of Zn was not applied to the control plants by Zn fertilizer?
Answer: We are grateful for the suggestion, however, in this manuscript we have decided to compare the effects of foliarlly applied ZnO-NPs to NPs free variant under field condition, similarly to our previous work by Kolenčík et al. (2019).
Comment #4 Results and discussion section needs substantial revision or rewriting. First, the results of the experiment must be presented/described. Second, discuss the novelty and perspective of the findings with relevant literature. This manuscript did not follow this sequence. For example, 3.2 subsection is written like a review article!
Answer: We agree, thus, the results and discussion sections were completely rewritten, and several new data were included, especially in 3.2 subsection.
Comment #5 Table 3 should be revised with units of the obtained data.
Answer: Since we do agree, the table was revised accordingly.
Comment #6 What is phosphor in Table 4? Why do all the elements in the lentil seeds decrease?
Answer: The information on phosphor was added to the manuscript as follows: “This balance in fully ripe lentil seeds is important for higher phytic acid content because this absorbs further micronutrients and contains phosphor as an antinutrient. Intestinal digestion further negatively affects nutrient bioavailability [27]”, and the reasoning for the decreasing trend of nutrients’ content was given :“While there were no statistically significant differences in the analyzed lentil seed nutrients for both variants, slightly higher average values were recorded for the control treatment”.
Comment #7 Figure 5 should not be included in the results section. ZnO NPs treated plants although the yield and physiology were improved.
Answer: There, we have placed the “environmental conditions”, including the newly added data on hours of sunshine associated with seasonal dynamics, intentionally because of the better interpretation of calculated physiological indices which pose a novelty of our manuscript.
References
Liu, H., Colombi, T., Jäck, O., Keller, T. and Weih, M. (2022) Effects of soil compaction on grain yield of wheat depend on weather conditions. Science of The Total Environment 807, 150763.
Kolenčík, M., Ernst, D., Komár, M., Urík, M., Šebesta, M., Dobročka, E., Černý, I., Illa, R., Kanike, R., Qian, Y., Feng, H., Orlová, D. and Kratošová, G. (2019) Effect of foliar spray application of zinc oxide nanoparticles on quantitative, nutritional, and physiological parameters of foxtail millet (Setaria italica L.) under field conditions. Nanomaterials 9(11), 1559.
Round 2
Reviewer 1 Report
The manuscript of Marek Kolenčík and co-authors has been significantly revised and supplemented. Nevertheless, I have some questions and comments again. The article must be revised again before publication in this journal.
1) What do the authors mean by “stress response” and “positive stress response”? Is the ZnO-NPs treatment stressful? Or stress is a result of changes in environmental conditions and NPs treatment results in plant resistance increase? In the abstract “Therefore, spray dispersion of 1 mg·L-1 ZnO-NPs was applied to lentil plants to examine the direct effects of NPs on its production, seed nutritional quality and stress response under field conditions.” In conclusion “The application of low NP concentrations has a significant impact on plant stress responses.” And from what the authors concluded that the plants were under stress? Please add clarifying information to the text.
2) Yet it is unclear from the article how nanoparticles work and whether it matters that they contain zinc oxide. And what is known about the conversion of zinc oxide when it enters plant tissue after foliar application.
3) Introduction “This balance is fully ripe lentil seeds is important for higher phytic acid content because this absorbs further micronutrients and contains phosphor as an antinutrient. Intestinal digestion further negatively affects nutrient bioavailability [27].” Please explain how and why intestinal digestion negatively affects nutrient bioavailability.
4) Designations a,b,c are absent in Figure 1. Figure 1 does not contain experimental data and should be transferred to additional materials.
5) It is not clear from the text of the article what dose of nanoparticles the plants received (only the concentration is given) and the degree of NPs penetration into plant tissues.
6) Please specify what is the “temperature difference”.
7) Table 3. “There was also a statistically significant difference confirmed for plant temperature and temperature difference in the ZnO-NPs variant, but not in untreated plants.” Significance is given for values for treated plants relative to untreated? Please clarify.
8) Table 4. “Table 4 lists plant micronutrients available from the extractable fraction of the experiment soil. All determined bioavailable Cu, Fe and Mn values correspond to results reported by Bloem, et al. [51]. Zinc is one exception because its higher content most likely originated from residues deposited in the soil after ZnO-NPs foliar application.” What does the given data refer to? At what point they were received? How do they differ from those for untreated plants?
9) Table 5. “The effects of foliar application of ZnO-NPs on content (g×kg-1) of selected essential and beneficial nutrients in lentil seeds after harvesting.” At the same time “There were no statistically significant differences in phosphor content as a seed antinutrient between the control and ZnO-NPs treated plants, and its value corresponded with the reported average [57].” Please clarify about phosphor.
10) “All differences in the analyzed seed nutrient contents of treated and untreated lentil plants are not statistically significant, but the average nutrient content values are slightly higher in the control variant. This may be due to the dilution effect, because the ZnO-NPs variant biomass yield was higher. A similar effect has also been noted by other authors [11, 61, 62].” Must be added for "most nutrients'" (for Potassium is not true). Moreover, it is not clear what the authors mean under dilution, since table 5 shows the data - g/kg (per plant mass?).
11) Figure 4. Designations a,b,c,d are absent. Significance is shown somewhere for control plants and somewhere for treated plants. How many plants have been counted to obtain this significance with overlapping standard deviations?
12) Figure 5. Designations a,b,c are absent. Figure 5 does not contain experimental data and can be transferred to additional materials.
13) “Further, the p = 0.0001 statistically significant difference was noted for all physiological indices (Figure 4) when the average air temperature was above 19°C (Figure 5a) and the plants were exposed to 10 hours minimum sunshine (Figure 5b).” Fig. 4 does not show the difference 0.0001. “Rainfall precipitation also appears partly responsible for observed effects. The statistically significant differences for all measured physiological indices were primarily noted when the average weekly precipitation was less than 18mm (Figure 5c). This is important because average weekly precipitation above 19mm decreases the atmospheric temperature, and this higher precipitation associated with slightly higher soil water content was confirmed by measurements on the 87th day. There was increased lentil transpiration, and no statistically significant differences were noted for any physiological indices at that time.” The significance of the differences is shown on day 87 in fig. 4a (Ig).
14) “Lentil is sensitive to photo-thermal conditions and its response to temperature and photoperiod is expressly linear [6].” What is meant?
15) “It is therefore highly likely that the positive stress response to NPs is enhanced when photo-thermal conditions overcome a critical value, and the lentil seed yield is then higher [6].” What critical values are we talking about?
16) “While there were no statistically significant differences in the analyzed lentil seed nutrients for both variants, slightly higher average values were recorded for the control treatment.” What two variants are we talking about? And yet, the significance of the differences for which values are given.
Author Response
Reviewer 1
Comment #1 What do the authors mean by “stress response” and “positive stress response”? Is the ZnO-NPs treatment stressful? Or stress is a result of changes in environmental conditions and NPs treatment results in plant resistance increase? In the abstract “Therefore, spray dispersion of 1 mg·L-1 ZnO-NPs was applied to lentil plants to examine the direct effects of NPs on its production, seed nutritional quality and stress response under field conditions.” In conclusion “The application of low NP concentrations has a significant impact on plant stress responses.” And from what the authors concluded that the plants were under stress? Please add clarifying information to the text.
Answer: This information has been added to manuscript as follows: “There stress is a result of changes in environmental conditions and NPs treatment results in plant resistance increase.”, and “There is then its additional influence in plant temperature and temperature variations which are useful indicators of plant transpiration intensity and energy balance covers various stress responses [26].”, Also, the term of “positive stress response” was changed to more logical formulation “improved stress response” for several parts of manuscript.
Comment #2 Yet it is unclear from the article how nanoparticles work and whether it matters that they contain zinc oxide. And what is known about the conversion of zinc oxide when it enters plant tissue after foliar application.
Answer: Information about zinc “Zinc is also an essential micro-nutrient in general metabolism [13, 14] and is a fundamental part of more than 300 enzymes [15], it is involved in photosynthesis, DNA replication and transcription [13], hydrocarbon metabolism [16], regulation of auxin-production [17] and cell membrane integrity [13]. While macro-sized and soluble-Zn-ion fertilizers are traditionally applied in agronomic practise [18], the ZnS [19] and ZnO nanoparticles are promising novel fertilizer nutrients for crops [11, 20, 21].”, and effect on ZnO nanoparticles with translocation to plant have been already evaluated “Nanoparticle metal oxides such as ZnO-NPs are increasingly applied as nano-fertilizers and agrochemicals to enhance crop production [21, 43, 44]. Liu and Lal [45] recorded that the advantage of nano-domains is their gradual release of beneficial nutrients, and this encourages the most effective plant growth and development. That release also induces photosynthesis, based on photocatalytic effect similar to TiO2-NPs [46, 47]. Singh, et al. [48] noted that ZnO-NPs’ positive effect on plant growth is achieved”, and translocation to plant were already described on manuscript…”, or “Transport velocity is also partly limited. However, ZnO-NP exposure to sunlight can initiate photo-corrosion which is usually associated with transformation to more bioavailable Zn2+ species. These ions are complexed to organic acids and transported to plant tissues through cuticle access [40].”.
Comment #3 Comment #9 Introduction “This balance is fully ripe lentil seeds is important for higher phytic acid content because this absorbs further micronutrients and contains phosphor as an antinutrient. Intestinal digestion further negatively affects nutrient bioavailability [27].” Please explain how and why intestinal digestion negatively affects nutrient bioavailability.
Answer: We appreciate the suggestion from the reviewer, however, the current manuscript studies the field experiment of lentil that is characteristic for the unpredictable weather conditions resulted to nutritional seed quality of lentil, not only biogeochemical cycles of phosphorus after intestinal digestion.
Comment #4 Comment #12 Designations a,b,c are absent in Figure 1. Figure 1 does not contain experimental data and should be transferred to additional materials.
Answer: Figure 1, Figure 5 shown the weather conditions during experimental season, and evaluation via LSD test is not standard for this type of info & graphs.
Comment #5 It is not clear from the text of the article what dose of nanoparticles the plants received (only the concentration is given) and the degree of NPs penetration into plant tissues.
Answer: This information has been added to manuscript as follows: “One liter of spray liquid was applied for each replication of the experiment.”
Comment #6 Please specify what is the “temperature difference”.
Answer: This information has been added to manuscript as follows: “The temperature difference is the difference between the plant temperature and the atmospheric temperature (Equation 1). If the temperature difference is positive or equal to zero, the plant is under stress while the negative value corresponds to better the plant's resistance to environmental stress. The lower temperature of the plant is caused by intensive transpiration.
T dif = T leaf – T air (1)”
Comment #7 Table 3. “There was also a statistically significant difference confirmed for plant temperature and temperature difference in the ZnO-NPs variant, but not in untreated plants.” Significance is given for values for treated plants relative to untreated? Please clarify.
Answer: This information has been added to manuscript with reference as follows: “…10 of same plants were measured in each replicate, and 10 measurements were made diagonally on each leaf according to the methodology of Jones et al. 2009.“ In total there are 300 values for one variant and one measurement (5,400 values for whole season).
Comment #8 Table 4. “Table 4 lists plant micronutrients available from the extractable fraction of the experiment soil. All determined bioavailable Cu, Fe and Mn values correspond to results reported by Bloem, et al. [51]. Zinc is one exception because its higher content most likely originated from residues deposited in the soil after ZnO-NPs foliar application.” What does the given data refer to? At what point they were received? How do they differ from those for untreated plants?
Answer: There were no significant difference on the extractable content of zinc in soil system observed between both variants after harvesting.
Comment #9 Table 5. “The effects of foliar application of ZnO-NPs on content (g×kg-1) of selected essential and beneficial nutrients in lentil seeds after harvesting.” At the same time “There were no statistically significant differences in phosphor content as a seed antinutrient between the control and ZnO-NPs treated plants, and its value corresponded with the reported average [57].” Please clarify about phosphor.
Answer: The phosphorus in lentil seeds as antinutrient depend on higher concentration range what was explained in intro section “This balance in fully ripe lentil seeds is important for higher phytic acid content because this absorbs further micronutrients and contains phosphor as an antinutrient. Intestinal digestion further negatively affects nutrient bioavailability [27].”, and there is no need to confuse potential readers in Tab. 5.
Comment #10 “All differences in the analyzed seed nutrient contents of treated and untreated lentil plants are not statistically significant, but the average nutrient content values are slightly higher in the control variant. This may be due to the dilution effect, because the ZnO-NPs variant biomass yield was higher. A similar effect has also been noted by other authors [11, 61, 62].” Must be added for "most nutrients'" (for Potassium is not true). Moreover, it is not clear what the authors mean under dilution, since table 5 shows the data - g/kg (per plant mass?).
Answer: This information has been added to manuscript as follows: “All differences in the analyzed seed nutrient contents of treated and untreated lentil plants are not statistically significant, but the average nutrient content values are slightly higher in the control variant expect of potassium.”, and “While there were no statistically significant differences in the analyzed lentil seed nutrients for both variants, slightly higher average values were recorded for the control variant expect of potassium.”;
The term dillution is common phenomenon in plant production, quantity versus quality. For more readable context we added to manuscript as follows: “This may be due to the dilution effect, because the ZnO-NPs variant biomass yield was higher, or improved lentil stress response.“;
For determination of seed mineral nutrients were added new reference “Losak, T., Hlusek, J., Martinec, J., Jandak, J., Szostkova, M., Filipcik, R., Manasek, J., Prokes, K., Peterka, J., Varga, L. and Ducsay, L., 2011. Nitrogen fertilization does not affect micronutrient uptake in grain maize (Zea mays L.). Acta Agriculturae Scandinavica, Section B-Soil & Plant Science, 61(6), pp.543-550.”
Comment #11 Figure 4. Designations a,b,c,d are absent. Significance is shown somewhere for control plants and somewhere for treated plants. How many plants have been counted to obtain this significance with overlapping standard deviations?
Answer: Please, check our response to Comment # 7
Comment #12 Figure 5. Designations a,b,c are absent. Figure 5 does not contain experimental data and can be transferred to additional materials.
Answer: Please, check our response to Comment #4
Comment #13 “Further, the p = 0.0001 statistically significant difference was noted for all physiological indices (Figure 4) when the average air temperature was above 19°C (Figure 5a) and the plants were exposed to 10 hours minimum sunshine (Figure 5b).” Fig. 4 does not show the difference 0.0001. “Rainfall precipitation also appears partly responsible for observed effects. The statistically significant differences for all measured physiological indices were primarily noted when the average weekly precipitation was less than 18mm (Figure 5c). This is important because average weekly precipitation above 19mm decreases the atmospheric temperature, and this higher precipitation associated with slightly higher soil water content was confirmed by measurements on the 87th day. There was increased lentil transpiration, and no statistically significant differences were noted for any physiological indices at that time.” The significance of the differences is shown on day 87 in fig. 4a (Ig).
Answer: This information has been rewritten, or added to manuscript with reference as follows: …“ Further, the p = 0.001 statistically significant”…, and “Rainfall precipitation also appears partly responsible for observed effects. The statistically significant differences for almost all measured physiological indices were primarily noted when the average weekly precipitation was less than 18mm (Figure 5c). This is important because weekly precipitation above 19mm decreases of average the atmospheric temperature, and this higher precipitation associated with slightly higher soil water content was confirmed by measurements on the 87th day. Trend of increased of lentil transpiration was observed, and no statistically significant differences expect one Ig physiological indices were noted at that time.”
Comment #14 “Lentil is sensitive to photo-thermal conditions and its response to temperature and photoperiod is expressly linear [6].” What is meant? Comment 15 “It is therefore highly likely that the positive stress response to NPs is enhanced when photo-thermal conditions overcome a critical value, and the lentil seed yield is then higher [6].” What critical values are we talking about?
This information has been added to manuscript with reference as follows: “Lentil is sensitive to photo-thermal conditions and its response to temperature and photoperiod is expressly linear [6]. It is therefore highly likely that the improved stress response to NPs is enhanced when photo-thermal conditions overcome a critical value of 10 hours photoperiodic lighting with corresponding temperature, and the lentil seed yield is then higher [6].„
Comment #16 “While there were no statistically significant differences in the analyzed lentil seed nutrients for both variants, slightly higher average values were recorded for the control treatment.” What two variants are we talking about? And yet, the significance of the differences for which values are given.
Answer: Information about two variants with level of statistical differences, concretely for ZnO-NPs and control (NPs free) have been already mentioned for whole manuscript e.g. “The experiment was performed on 5m2 parcels with silt loam soil [11]. Three parallel runs were made for each variant, and two variants were settled in random orientation in the perpendicular blocks”, …“Table 3”…, “…the nitrogen concentration in both variants corresponded to the typical nitrogen content in lentil seeds…” etc.
Reviewer 2 Report
The revised version is acceptable for publication.
Author Response
Comments and suggestions of reviewer were evaluated in manuscript.
Round 3
Reviewer 1 Report
Accept in present form.
Author Response
Comment #1 ZnO is a highly published material and in that respect, the introduction should be detailed in a way that it reflects the adequate literature background. Authors can look into relevant literature, for example, Materials Today 21, 2018, 631-651; NPG Asia Materials 11, 2019, 1-13; and many others.
Answer: This information has been added to manuscript with reference as follows “…Therefore, its morphological, spatial and structural forms in glass, electronics, energy, pigments, rubber textiles, cosmetics, food additives and pharmaceutics and medicine are complemented by its use in other technology [22, 24, 25]…”
Comment #2 When it comes to ZnO nanoparticles, at least readers expect to see some morphological (SEM, TEM/HRTEM, ...) and structural (XRD, Raman), optical (PL, ....), etc. properties, in the manuscript. It will increase the visibility and the reachability of the paper to a broad audience.
Answer: Scanning transmission electron microscopy (STEM) of ZnO nanoparticles with bright and dark modes & X-ray diffraction analysis have been added to manuscript as Figure 3a, and Figure 3b.
Comment #3 Since there is no color charge as the paper is only going to be published online, why not making the colored bar charts in figure 1 and also the legends need to be improved, the first letter could be capital, like, 'Precipitation'...
Answer: This information has been added/ changed according to editor suggestion.
Comment #4. Some more discussion about the enhanced growth mechanism could be briefly highlighted.
Answer: This information has been added to manuscript with reference as follows “…Our results are supported by the foliar application of 20 mg×L-1 ZnO-NPs to Abelmoschus esculentus which increased its pod number and improved plant growth and yield [53]…” or “…Zinc is required by plants for carbohydrate metabolism and gene expression in response to environmental stress [46], and ZnO-NPs…”
Comment #5. The manuscript still needs to be improved in terms of English level and citations.
Answer: English language corrections involved in manuscript was checked by English lector Dr. Raymond Joseph Marshall.